# Evaluation of Diagnostic Performance of Automatic Breast Volume Scanner Compared to Handheld Ultrasound on Different Breast Lesions: A Systematic Review

**DOI:** 10.3390/diagnostics12020541

**Published:** 2022-02-19

**Authors:** Shahad A. Ibraheem, Rozi Mahmud, Suraini Mohamad Saini, Hasyma Abu Hassan, Aysar Sabah Keiteb, Ahmed M. Dirie

**Affiliations:** 1Department of Radiology, Faculty of Medicine and Health Sciences, Universiti Putra Malaysia, Serdang 43400, Malaysia; rozi@upm.edu.my (R.M.); surainims@upm.edu.my (S.M.S.); hasyma@upm.edu.my (H.A.H.); 2Centre for Diagnostic Nuclear Imaging, Universiti Putra Malaysia, Serdang 43400, Malaysia; 3Department of Radiological Techniques, College of Health and Medical Technologies, Baghdad 10047, Iraq; aysarph.d@gmail.com; 4Department of Community Health, Faculty of Medicine and Health Sciences, Universiti Putra Malaysia, Serdang 43400, Malaysia; diiriyenih@gmail.com

**Keywords:** handheld ultrasound, BI-RADS, automatic breast volume scanner, breast cancer

## Abstract

Objective: To compare the diagnostic performance of the automatic breast volume scanner (ABVS) against the handheld ultrasound (HHUS) in the differential diagnosis of benign and malignant breast lesions. Methods: A systematic search and review of studies involving ABVS and HHUS for breast cancer screening were performed. The search involved the data taken from Scopus, PubMed, and science direct databases and was conducted between the year 2011 to 2020. The prospective method was used in determining the inclusion and exclusion criteria while the evidence level was determined using the BI-RADS categories for diagnostic studies. In addition, the parameters of specificity, mean age, sensitivity, tumor number, and diagnostic accuracy of the ABVS and HHUS were summarized. Results: No systematic review or randomized controlled trial were identified in the systematic search while one cross-sectional study, eight retrospective studies, and 10 prospective studies were found. Sufficient follow-up of the subjects with benign and malignant findings were made only in 10 studies, in which only two had used ABVS and HHUS after performing mammographic screening and MRI. Analysis was made of 21 studies, which included 5448 lesions (4074 benign and 1374 malignant) taken from 6009 patients. The range of sensitivity was (0.72–1.0) for ABVS and (0.62–1.0) for HHUS; the specificity range was (0.52–0.98)% for ABVS and (0.49–0.99)% for HHUS. The accuracy range among the 11 studies was (80–99)% and (59–98)% for the HHUS and ABVS, respectively. The identified tumors had a mean size of 2.1 cm, and the detected cancers had a mean percentage of 94% (81–100)% in comparison to the non-cancer in all studies. Conclusions: The evidence available in the literature points to the fact that the diagnostic performance of both ABVS and HHUS are similar with reference to the differentiation of malignant and benign breast lesions.

## 1. Introduction

Breast disease is common among modern women. It is also one of the leading diseases that threatens the physical health of women. The American Cancer Society predicted that the United States would top the ranking, with 276,480 women with breast cancer in 2020 in the country alone, which accounted for 30% of all cancer patients. With the death of 42,170 of these women, it ranked in second place by accounting for 15% of cancer deaths [1]. In Malaysia, breast cancer is the leading type of cancer, which accounted for 34.1% of all cancer cases in the female population. The diagnosis of a total of 21,634 female breast cancer cases were made between 2012 and 2016, compared to 18,206 cases in the report of 2007–2011. Nonetheless, 19.0% of all new diagnosis of cancer cases between 2012–2016 in comparison to 17.7% in 2007–2011 was attributed to breast cancer, in spite of gender. There has been an increase of 2% for overall cancer among women in a similar comparative period, from 32.1% to 34.1%, for new cases of breast cancer [2].

However, the ideal breast screening system is yet undetermined. Mammography has for some time been the only suggestion for breast screening imaging assessment, and has shown brilliant affectability and particularity. Nonetheless, its utilization in young women under 40 years of age is restricted due to radiation concerns and, often, thick breasts. Presently, there has been a decline in the once-demonstrated power of yearly mammography screening in decreasing the rate of mortality related to breast cancer [3]. The overall clinical limitation of breast screening is similarity in magnetic resonance imaging (MRI) due to inadequate particularity and inadmissible accessibility to non-wealthy regions and individuals [4].

Apart from the obvious benefit of ultrasound in breast cancer diagnosis, it is also beneficial in complementing diagnostic mammograms. It is safe and sensitive at distinguishing the echo of gland tissue and fat, and is good at defining the boundary and morphology of lesions [5,6,7]. Despite the benefits, the duration of the whole-breast examination and operator dependence has rendered the conventional handheld ultrasound (HHUS) with a number of inherent limitations [8]. In contrast, shorter duration of image acquisition, higher reproducibility, and less operator dependence of the automated breast volume scanner (ABVS) has given it a number of advantages over the HHUS [9,10]. Besides, extra information on the diagnosis of the coronal plane that has been reconstructed can also be obtained from the ABVS. Hence, these advantages render the ABVS a promising method for the imaging of breasts in screening as well as diagnostic settings [11,12]. ABUS offers reproducible, high-resolution images and does not depend on the operator, and is achieved using an automated scanner with a larger field of view. Numerous prospective studies have described that adding mammography to ABUS screening resulted in similar positive outcomes to those linked with HHUS screening, such as increased discovery of invasive cancer and reduced rates of interval cancer [5].

In 2012, the approval for the use of ABVS was granted by the U.S. Food and Drug Administration for women with dense breasts and negative mammography findings as an additional whole-breast screening method [13]. The use of ABVS for diagnostic purposes has, over the past decade, been examined by different studies [9,14,15,16,17]. Promising results have been reported in several considerably small patient-population studies [15,18,19,20]. However, the differentiation and characterization of breast lesion discovered through mammography or other screening technologies in present day clinical practice are conducted by a majority of radiologists with the use of conventional HHUS. The controversy regarding the diagnostic performance of ABVS in contrast to HHUS remains. Hence, a systematic review of the differential diagnosis of breast lesions was conducted, comparing HHUS and ABVS diagnostic performance.

## 2. Materials and Methods

### 2.1. Search Strategy

A strategy was developed for a systematic search to distinguish significant literature. The search strategy was customized to three databases, namely PubMed, Scopus, and science direct, while the inquiry terms ‘automated breast ultrasound’, ‘handheld ultrasound’, and ‘breast cancer’ were utilized. In particular, all literature was included with reference to automated breast ultrasound (which included its synonyms, for example, automated breast sonography, automated breast scanner, automated whole breast ultrasound, automated breast image, automated whole breast volume scan, 3D automated breast ultrasound, automated breast volume ultrasonography) and handheld (handheld, handheld, handheld, portable, or pocket). The search also spanned the period from the inception of the database to 2020, with the inclusion of journal articles published solely in English.

### 2.2. Selection Criteria

The criteria for selection depended on the PRISMA Statement [21]. The search mainly focused on the mapping of existing literature on automated breast ultrasound in contrast to handheld ultrasound in medicine, biochemistry, genetics, molecular biology, and the health professions. The search then narrowed down to the medical field. The search span was made between 2011–2020. All articles before 2011 were excluded. The search was not limited to any specific countries, therefore there was no exclusion in this option. The following are the inclusion criteria: (1) both ABVS and HHUS were used in breast lesion diagnosis; (2) the ABVS method was financially accessible; (3) the study population was made up of a minimum of 20 patients; (4) follow-up of histologic analysis (surgery or biopsy), and clinical/imaging for a minimum of 1 year, and unchanged lesions were viewed as pathologically benign. Screening of the relevant literature’s abstracts and titles were conducted, and inspection of the full texts was performed by two researchers independently in determining the inclusion of selected articles in the analysis. A consensus was used in resolving any conflict between the two researchers. At this stage, a total of 311 research articles were excluded while 414 records were extricated.

### 2.3. Data Extraction

Data collection was made regarding the year of publication, the country in which the examination was performed, the objectives, study design, number of participants, screening methods of assessment, patients’ mean age, and the number of lesions.

### 2.4. Quality Assessment

Conference papers, original research articles, and review papers became the basis of this study. A thorough check was made on all duplications in maintaining the nature of the review. In ensuring the relevance and quality of the academic literature included in the review process, detailed examination of the abstracts of articles was conducted in the process of analysis and purification. Next, careful assessment of each research paper was performed. To limit the research only to English-published papers, the subsequent exclusion criterion was therefore used. Therefore, three articles in languages other than English have not been included in the study. In addition to that, the filtration of duplicate records resulted in the removal of 65 more articles. The assessment of each article based on the inclusion and exclusion criteria above resulted in the selection of 21 articles. The exclusion and inclusion of the literature at every stage (PRISMA Statement) is shown in Figure 1. Assessment of the methodological quality was made by two independent reviewers, and to resolve any dispute between the reviewers, mutual suggestion was used. For the inclusion of studies regarding ‘diagnostic accuracy’, the QUADAS-2 (Quality Assessment of Diagnostic Accuracy Studies-2) tool was used which involved four domains including ‘index test’, ‘reference standard’, ‘patient selection’, and ‘flow and timing’ [22] (Table 1). Evaluation was made of each domain with regard to its risk of bias (low, high, or unclear), and the initial three domains were identified with regard to their applicability. In general, a study that is viewed as “low” in all domains regarding its applicability or bias is deemed appropriate for an overall judgment of “low concern regarding applicability” or “low risk of bias” for the study. However, judgement of a study is made as having “concerns regarding applicability” or “at risk of bias” if viewed as “unclear” or “high” in one or more domains. Dispute between the two reviewers in assessing the quality of the study was settled through discussion (Figure 2).

## 3. Results

### 3.1. Search Strategy and Study Selection

Based on the search strategy, 726 records were discovered from the electronic databases when, due to duplication issues, 311 items were then discarded. The screening of titles and abstracts based on the inclusion criteria resulted in the exclusion of another 320 items. By reading the full content, 64 more items were excluded based on the examination of the remaining 85 articles. The final selection to be included in the literature review of this study included 21 studies (Depretto et al., 2020; Jia et al., 2020; Tutar et al., 2020; Yun et al., 2019; Zhang et al., 2019; Niu et al., 2019; Choi et al., 2018; Zhang et al., 2018; Schmachtenberg et al., 2017; Hellgren et al., 2016; Kim et al., 2016; Jeh et al., 2015; Choi et al., 2014; Chen et al., 2013; Kim et al., 2013; Lin et al., 2012; Wang et al., 2012; Wojcinski et al., 2011; Chang et al., 2011; Shin et al., 2011). The screening of the included studies’ reference list marks the end of the process of inclusion of studies. A summary of the process involved in the studies selected for the inclusion of this review is shown in the flowchart (see Figure 2).

### 3.2. Characteristics of the Study and Quality Assessment

From the 21 studies, the inclusion of 5448 lesions in 6009 patients was made, whereby 1374 (23.3%) were malignant while 4074 (69.1%) were benign. Three brands of the ABVS were discovered; ACUSON S2000 (Siemens Healthcare, Erlangen, Germany) as the highest-utilized brand among 13 studies, InveniaTM (GE Healthcare, WI, USA) as the second highest utilized brand among five studies, and SomoVu Scan Station (Usystem, Inc., San Jose, CA, USA), found in three studies although five brands of the HHUS were discovered; ACUSON S2000™; Siemens, Erlangen, Germany as the highest-utilized brand among 8 studies, iU22 Ultrasound System (Philips Medical System, WA, USA) as the second highest utilized brand among 7 studies, Logiq E9 (GE Healthcare; Milwaukee, WI, USA) utilized among three studies, EUB-8500 scanner (Hitachi Medical, Tokyo, Japan) used in two study while Aplio 80 (Toshiba, Tokyo, Japan) just in one study.

Among 14 studies, the sensitivity range was (0.72–1.0) for ABVS and (0.62–1.0) for HHUS, while the specificity range was (0.52–0.98) for ABVS and (0.49–0.99) for HHUS. Among the 11 studies, the accuracy range was (59–98)% and (80–99)% for ABVS and HHUS, respectively. ABVS and HHUS were utilized in 21 studies, thus giving unbiased, extractable data in terms of diagnostic accuracy. For the 21 studies on ‘diagnostic accuracy’, the results of histopathology assessment were the standard of reference. ‘Lesion by lesion’ is the rater approach chosen in all studies. The raters evaluated each lesion detected for malignancy using the BI-RADS. Table 1 provides the summary of the details of the study characteristics. As demonstrated by the QUADAS-2 tool, most of the studies (15 out of 21) had a rather high methodological quality (Table 2). However, due to the use of case-control configuration in this study, one study was judged as high risk [23]. As a result of imprecise review of HHUS and ABVS images, the same radiologist judged five studies as unclear risk, with the use of the blinding method.

### 3.3. Age Distribution

The report of 18 studies showed an age range between 11 to 82 years old, with the overall age range exceeding 30 years old in every study. Information on the median age was provided in three studies (49, 49, and 52 years), with age range between 32 to 82 years [24,25,26] (Table 2).

### 3.4. Breast Lesion Diagnosis Using the Categories of BI-RADS

Two studies separated the HHUS and ABVS outcome of breast screening according to the BI-RADS 4 and BI-RADS 5 classification [25,27]. Other studies assessed the results of breast tissue according to BI-RADS 1–BI-RADS 5. Females with BI-RADS 4 and BI-RADS 5 breast categories demonstrated the largest proportion of breast cancer diagnosed through ultrasound screening. Depretto et al. analyzed four carcinomas distinguished by (BI-RADS 4) breast tissue, and 130 cases with breasts in categories (BI-RADS 1 and BI-RADS 2) [28], and 29 malignancies were diagnosed in type 4 and type 5 BI-RADS breast tissue in Jia et al., while two carcinomas were found in three breasts [23].

Using the ABVS, the discovery of 51 circumscribed solid nodules (BI-RADS 3) were made in 42 women. The HHUS exhibited five of these as complicated cysts. The primary HHUS missed five nodules. However, the detection of five BI-RADS 3 solid nodules, one BI-RADS 2 implant rupture, and one BI-RADS 4 distortion was made by HHUS, which were all missed by ABVS. Altogether, 78 lesions were found in 340 women, 71 (91%) of the detected nodules were discovered by ABVS while 68 (87.2%) of the detection was made by the primary HHUS [24]. Niu et al. included 599 masses detected in 398 women (which included solid and cystic masses). The classification of 359 masses by the HHUS and ABUS as category 2 or 3 indicated these as benign masses. The MRI classified two of these masses as category 4 or 5 [29].

It was found at the completion of the study that 496 (83%) of the 599 masses were benign while the remaining 103 (17%) were malignant, with the one-year follow-up information or pathological outcomes as the reference standards. As indicated by the BI-RADS classification for every unit, 258 units (258/320, 80.63%) had BI-RADS classification 1–2, 62 units (19.38%) had BI-RADS classification 3 while 155 lesions had the introductory BI-RADS classifications of 4–5 [30]. Choi et al. confirmed 184 malignant cases of BI-RADS 4 and BI-RADS 5 classes, and 234 lesions were diagnosed as benign (BI-RADS 3) [31]. The remaining 413 lesions were assigned as BI-RADS class 3 (*n* = 292) or 2 (*n* = 121). In a study conducted by Zhang et al., 1353 females (68.6%) were characterized as BI-RADS 1, 2, or 3 categories while the other 620 females (31.4%) were classified as BI-RADS category 4 or 5 [32].

Jeh et al. reported the BI-RADS final assessment of 124 classifications as 1, 2, and 3 while 45 malignant classifications were 4 and 5 [33]. Lin et al. identified 15 carcinomas noticed in BI-RADS classification 4–5, and 20 females with breasts in classification 1-3 [34]. Wojcinski et al. identified that 6 out of 14 lesions had been classified as BI-RADS 0, 3 or 4 while 8 out of 14 malignant lesions had been properly classified as BI-RADS 5 [26]. Of masses that were benign, 39 were categorized as BI-RADS 4 while another 7 were characterized as BI-RADS 3. Of masses that were malignant, 16 were accounted for as BI-RADS 5, while 8 masses that were malignant were categorized as BI-RADS 4 [20]. The evaluation of 145 lesions in Shin et al. marked the final evaluation of the BI-RADS classes, where 145 lesions were accounted for by five readers as category 1 or 2 (40%, 240 of 603), 3 (31%, 184 of 603), 4A (11%, 68 of 603), 4B (3%, 19 of 603), 4C (2%, 12 of 603), and 5 (13%, 80 of 603) [35].

### 3.5. Tumour Stage and Size (Lymph Node and Non-Invasive/Invasive Status)

The detected carcinomas had a mean size of 2.1 cm (mean size ranged between 1.6 cm to 2.6 cm) in seven of the 21 studies [33,34,35,36,37,38,39]. The mean level of malignancy detected was 94% (81–100%) contrasted with non-cancer in all examinations. The status of intramammary lymph node was accounted for in one analysis, with lymph nodes that were negative in all studies [31] (Table 2).

### 3.6. Assessment Categories for Breast Ultrasound

The positive predictive value concerning the finding of malignancies in biopsies that are prompted/detected by ultrasound was accounted for, or had the option to be derived from the information provided by eleven studies. The percentage of positively-classified findings for carcinoma was then found to range between 4.1–100% for both ABVS and HHUS. The large variation of these positive predictive values is chiefly due to the application of different assessment categories and different sonographic criteria for malignancy. Only three studies reported that all findings were classified as benign post biopsy [35,39,40]. Nonetheless, three studies did not specify the follow-up for patients with positive outcomes [26,33,41] (Table 2).

## 4. Discussion

ABVS is a novel imaging method for automated breast scanning via ultrasound. The use of this method was first made in the screening setting to enhance the detection of breast cancer [5]. Recently, the evaluation of the use of ABVS in the diagnostic stage has been made by various studies. However, the diagnostic capability of the ABVS continues to be disputable since it is a “novel” method, especially in comparison with the traditional HHUS. It was reported in Meng et al. that ABVS demonstrated decreased specificity compared to HHUS, although both ABVS and HHUS demonstrated equal sensitivity [19]. In a meta-analysis in Zhang et al., the same detection rate (100%) was found in both HHUS and ABVS in detecting breast cancer. However, ABVS demonstrated numerically greater specificity (86% vs. 82%) and sensitivity (93% vs. 90%) in comparison with HHUS [30]. On the other hand, Wang and Qi reported that ABVS and HHUS demonstrated similar specificity and sensitivity in discriminating benign breast lesions from malignant. The pooled values of specificity and sensitivity for ABVS were 82.2% and 90.8%, respectively, while HHUS pooled specificity and sensitivity values were 81.0% and 90.6%, respectively [42].

This systematic review aimed to determine the evidence available on ABVS and HHUS diagnostic accuracy in identifying malignant and benign breast lesions. Based on the data in this study, similar sensitivity and specificity were demonstrated for ABVS and HHUS in discriminating malignant and benign breast lesions. Among 14 studies, the sensitivity range was (0.72–1.0) for ABVS and (0.62–1.0) for HHUS and the specificity range was (0.52–0.98) for ABVS and (0.49–0.99) for HHUS; while among 11 studies, the accuracy range was (59–98)% and (80–99)% for ABVS and HHUS, respectively. On the other hand, as suggested in the studies of Kim et al. and Wang et al., ABVS can be effectively utilized in detecting and characterizing breast lesions, since its sensitivity is not inferior [36,40]. As found in the study of Tutar et al., ABVS succeeded in detecting all malignant lesions, apart from the fact that no interval cancers were detected in the very long follow-up period. A greater number of benign lesions could be fundamentally identified by ABVS in comparison to HHUS. There is a possibility that a proportion of solid nodules detected using ABVS could be focal fat lobules [24].

The article review has shown that a statistically significant difference in results exists between HHUS and ABVS, since HHUS contained a larger amount of BI-RADS 1–2 compared to BI-RADS 0 or 4 discovered in ABVS. This indicates the probability of ABVS resulting in better clinical practice regarding biopsies, follow-ups, and recalls, in comparison to HHUS.

Based on the outcomes of Yun et al. it is confirmed that a relationship exists between ABVS allocation of a lower BI-RADS category and HHUS allocation of US results of irregular shape [25].

Posterior shadowing is a notable limitation of ABVS, identified by the recall or false-negative rate. Notwithstanding the findings for lesion size, no association was revealed in our study group between orientation, margin, or posterior acoustic features and a lower categorization utilizing ABUS. It was shown in the study results of Zhang et al. that the ABVS findings were in greater sync with the pathological outcomes of the BI-RADS 4–5 groups [32]. The 78.6% of females diagnosed with cancer or precancerous lesions, from those classified by ABVS as BI-RADS 4 or 5, was 7.2% greater compared to the female proportion in a similar group of BI-RADS classification based on HHUS. However, the false-negative rates of the BI-RADS 1–2 groups for both ABVS and HHUS were not distinguishable from each other.

On the other hand, emphasis should be made on the fact that HHUS and ABVS diagnostic performance similarity is largely dependent on the interpretation of the Gray-scale ultrasound. No clear statement on the usage of elastography and Doppler ultrasound in HHUS was made in any of the 21 studies included. This suggests that, for breast lesion differential diagnosis, no additional information on vascularity and elasticity was provided, when only the morphological features on Gray-scale images for HHUS and ABVS were used. Breast ultrasound diagnostic accuracy could therefore be significantly improved due to the ability of both elastography and Doppler ultrasound to provide independent diagnostic information apart from Gray-scale imaging. Nevertheless, the investigation of tissue elasticity and lesion vascularity continues currently to be performed by ABVS in the clinical environment. Henceforth, the diagnostic performance of HHUS in the involved studies may be underestimated relative to the clinical reality. Therefore, there is a probability of an underestimation of HHUS diagnostic performance made in the involved studies in relation to clinical reality.

However, a great advantage of ABVS in breast lesion characterization in comparison to HHUS is its capability in obtaining additional data on the reconstructed coronal plane’s morphological features. In the differentiation of breast lesions that are malignant and benign, the ABVS coronal plane retraction phenomenon is perceived as having high probability as a diagnostic feature (Depretto et al., 2020; Jia et al., 2020; Tutar et al., 2020; Yun et al., 2019; Zhang et al., 2019; Niu et al., 2019; Choi et al., 2018; Zhang et al., 2018; Schmachtenberg et al., 2017; Hellgren et al., 2017; Jeh et al., 2015; Choi et al., 2014; Chen et al., 2013; Kim et al., 2013; Lin et al., 2012; Wang et al., 2012; Wojcinski et al., 2011; Chang et al., 2011; Shin et al., 2011). Thus, it can be sensibly concluded that in terms of differential findings assisted by coronal reconstruction, ABVS might be better when compared to HHUS.

Nonetheless, the reviewed article demonstrates that the diagnostic performance of ABVS is similar to the diagnostic performance of HHUS. In terms of benign and malignant breast lesion differential diagnosis, no confirmation was made in our study regarding the added benefit of using ABVS for coronal reconstruction. The results of Zhang et al. are in agreement with those of our study which indicated that, regarding AUC values, ABVS diagnostic performance might not be significantly improved through coronal reconstruction [30]. In the study of Zhang et al., a suggestion was made that ABVS independent value limitation in the differential diagnosis is caused by the low sensitivity (37.0%) of the retraction phenomenon on the coronal plane [32]. The limitations in our study include the substantial dominance of Asian reports, with 15 out of 21.

Variations might have occurred due to the uneven geographic distribution since there are breast cancer differences regarding region and ethnicity between non-Asian and Asian women. Besides, no indication was made in any of the studies included regarding an image quality control statement, which should thus be noted as a variable that is unaccounted for, both for ABVS and HHUS. Third, based on our references, no investigation has made use of Doppler and elastography ultrasound in HHUS, in contrast to the practice in clinical reality. Finally, publication bias might have been prompted since evaluation was made only of articles written in English. Therefore, our reviewed articles may have underestimated the diagnostic performance of HHUS.

## 5. Conclusions

In relation to malignant and benign breast lesion differentiation, ABVS diagnostic performance based on the evidence available in the literature is similar to that of HHUS. However, ABVS can offer new diagnostic information. ABVS may help to distinguish between real lesions. This technique is feasible for clinical applications and is a promising modality in breast imaging. Nevertheless, since this review of articles was conducted on various studies, most of which were obtained from a single geographical region, further studies are hence required before the generalization of this conclusion can be made. More sound research associating the diagnostic performance of ABVS and mammography/MRI is anticipated and required.

## Figures and Tables

**Figure 1 diagnostics-12-00541-f001:**
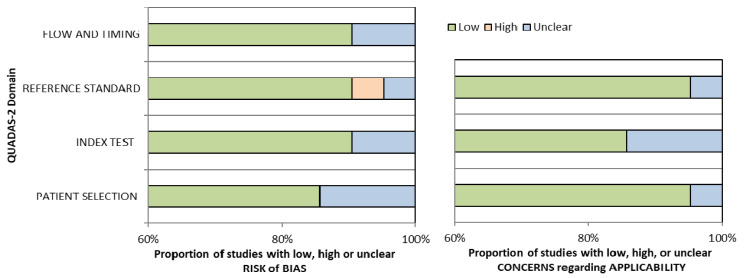
Bar charts for Quality Assessment of Diagnostic Accuracy Studies-2 (QUADAS-2) analysis for 21 studies of diagnostic accuracy.

**Figure 2 diagnostics-12-00541-f002:**
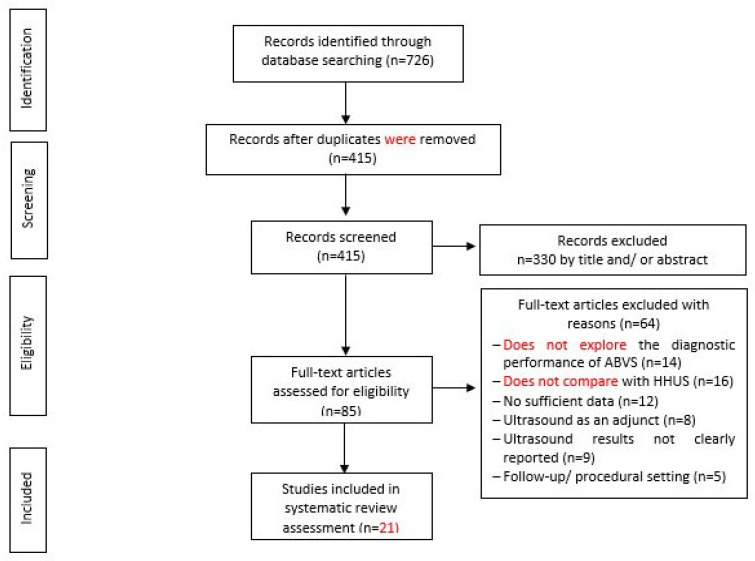
Flow Chart of Study Selection.

**Table 1 diagnostics-12-00541-t001:** Quality of studies included in diagnostic accuracy analysis for risk of bias and applicability concerns.

Study	Risk Of Bias	Applicability Concerns
Patient Selection	Index Test	Reference Standard	Flow and Timing	Patient Selection	Index Test	Reference Standard
Depretto et al., 2020							
Jia et al., 2020	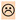			?			
Tutar et al., 2020				?			
Yun et al., 2019							
Zhang et al., 2019	?						
Niu et al., 2019							
Choi et al., 2018							
Zhang et al., 2018							
Schmachtenberg et al., 2017							
Hellgren et al., 2016							
Kim et al., 2016	?	?				?	
Jeh al., 2015			?				?
Chio et al., 2014					?		
Kim et al., 2013		?				?	
Chen et al., 2013	?						
Lin et al., 2012						?	
Wang et al., 2012							
Wojcinski et al., 2011							
Chang et al., 2011							
Shin et al., 2011							


 Low Risk 
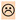
 High Risk ? Unclear Risk.

**Table 2 diagnostics-12-00541-t002:** Characteristics of the included studies.

Author	Study Design/Objectives/Participants	Screening Method	Findings	Outcome
Tutar et al., 2020 Turkey	**Study design** = prospective study **Objective** To prospectively compare between the automated breast volumetric scanning (ABVS) with the hand-held bilateral whole breast ultrasound (HHUS) in lesion detection, and characterization. **Participants** = 345 women **Age range** = between 35 to 67 years old (median 49-year-old) **Variables** = Breast cancer, Breast ultrasonography, Automated ultrasound, Mass screening.	**Screening device** = ➢ ABVS with integrated 14L5BV linear transducer (15.4 cm) ➢ HHUS with 14L5 (5–14 MHz) or 9L4 (4–9 MHz) linear transducer **Duration** = Between May 2014 and July 2015 **Follow up** = minimum 36 months **Interpreting Image** = BI-RADS US lexicon	➢ Recall rate was 46/340 (13.05%) for ABVS, and 4/340 (1.18%) for HHUS. ➢ HHUS had higher results of true negatives (BI-RADS 1–2) while ABVS had higher results of false positives (*p* < 0.001). ➢ ABVS had a positive predictive value of 4.17% while HHUS had 50%. ➢ In comparison to HHUS, ABVS had irregular nodules of (*p* < 0.001), distortions of (*p* < 0.034), and over diagnosed shadowing of (*p* < 0.01). ➢ 59.7% of the women mentioned that if they had a choice, they would have chosen HHUS. 10.6% of the women experienced severe pain from the use of ABVS.	✓ The use of ABVS in lesion detection is as good as HHUS. ✓ ABVS had a higher recall rate and lower positive predictive value. This could end up in greater follow-ups, and greater anxiety for the women. ✓ Had they been given the choice, more than 50% of the women would have preferred HHUS.
Depretto et al., 2020 Italy	**Study design** = Retrospective study **Objective** To examine the agreement between the hand-held ultrasound (HHUS) and the automated breast ultrasound (ABUS) in monitoring of women with breast cancer history, with regard to the contralateral breast cancer or recurrences or new ipsilateral **Participants** = 154 women **Age range** = aged between 34–90 years old (mean ± SD 62 ± 11 years) **Variables** = Diagnostic imaging, Breast oncology, Epidemiology, and Prevention	**Screening device** = ➢ The Selenia Dimension mammography system was used in conducting the mammography (Hologic; Bedford, MA). ➢ ABVS with a 6–14 MHz frequency. ➢ HHUS with a linear 6–15 MHz transducer. **Duration** = Between April to June 2016 **Follow up** = 18 to 24 months **Interpreting Image** = BI-RADS US lexicon	➢ ABUS and HHUS were substantial for dichotomic assessment (κ = 0.794) and for BI-RADS categories (κ = 0.785). ➢ There was a significant difference in assigning the BI-RADS categories (*p* < 0.05), although there was no difference in dichotomic assignment between 2 readers (*p* = 0.5).	✓ A substantial agreement was achieved between ABUS and HHUS in monitoring of women with breast cancer history. ✓ In particular, the ABUS could be used in first-level monitoring of intermediate risk women since it could recognize all cancers detected by HHUS.
Jia et al., 2020 China	**Study design** = Cross-sectional **Objective** To determine, both in combination with mammography and separately, the diagnostic performance of the automated breast ultrasound system (ABUS) and the handheld ultrasound (HHUS) in dense breasts Chinese women. **Participants** = 1973 women **Age range** = Between 30–69 years, mean age 49.1 years (SD: 6.8) **Variables** = Breast density, Mammography, Breast neoplasms, Ultrasonography.	**Screening device** = ➢ The mammograms were acquired using the Fujifilm FDR MS-2500 (Fujifilm Crop., Tokyo, Japan), GE Sengraphe DS (GE Healthcare, WI, USA), and Hologic Selenia (Hologic, MA, USA). ➢ The HHUS images were obtained using the Aixplorer system (Supersonic Imagine, Aix-en-Provence, France), GE LOGIQ9 (GE Healthcare, WI, USA), iU22 Ultrasound System (Philips Medical System, WA, USA), and S2000 (Siemens Medical Solutions, CA, USA). ➢ All ABUS scans were acquired using the Invenia ABUS (GE Healthcare, WI, USA). **Duration** = Between February 2016 and March 2017 **Follow up** = NP **Interpreting Image** = BI-RADS US lexicon	➢ In mammography-negative dense breasts, the rate for incremental cancer detection was 42.8 for each 1000 ultrasound examinations. ➢ The combination of HHUS or ABUS with mammography produced a sensitivity of 99.1% (219/221), and the specificities were 84.9% (608/716) and 86.9% (622/716), respectively. ➢ The combination of the HHUS with mammography produced a 0.92 area under the curve, while a combination of ABUS with mammography produced one at 0.93. ➢ An agreement that is statistically significant in breast cancer detection between HHUS and ABUS was observed (percent agreement = 0.94, κ = 0.85).	✓ As adjuncts to mammography, both the HHUS and ABUS can substantially increase the rate of breast cancer detection in dense breasts women; a strong correlation exists between them. ✓ With various benefits of the ABUS over HHUS, for instance reproducibility and less operator dependence, and the commonness of dense breasts, the use of ABUS in the early detection of breast cancer, especially in areas that have limited resources, has shown great potential.
Yun et al., 2019 Korea	**Study design** = Retrospective **Objectives** In terms of the assessment of Breast Imaging Reporting and Data System (BI-RADS) category—to assess the reliability of suspicious breast masses examination performed using the automated breast ultrasound (ABUS) as opposed to the handheld breast ultrasound (HHUS). To examine factors that affect categorization discrepancies. **Participants** = 135 patients **Age range** = between 34–90 years old (mean ± SD 62 ± 11 years) **Variables** = Breast Imaging Reporting and Data System, Breast neoplasms, Automated breast ultrasound, Hand-held ultrasound	**Screening device** = ➢ ABUS exams ➢ HHUS images were acquired using the linear transducer at 7–15 MHz **Duration** = Between July 2016 and December 2016 **Follow up** = NP **Interpreting Image** = BI-RADS US lexicon	➢ There was an overall good agreement in all cases between HHUS and ABUS (79.3%, kappa = 0.61, *p* < 0.001). ➢ It was revealed in the logistic regression analysis that differences in the categorization of BI-RADS were associated with the suspicious presence of microcalcification on the mammography (odds ratio [OR], 4.63; 95% confidence interval (CI), 1.83 to 11.71; *p* = 0.001) and an irregular shape on US (OR, 5.59; 95% CI, 1.43 to 21.83; *p* = 0.013).	✓ The examination of suspicious breast masses under the categorization of BI-RADS have demonstrated good agreement between HHUS and ABUS. ✓ The presence of an irregular shape on US and the accompaniment of suspicious microcalcifications on mammography were factors linked to the yielding of a lower level of suspicion in the ABUS compared to the HHUS regarding the assessment of BI-RADS category.
Zhang et al., 2019 China	**Study design** = prospective study **Objective** To investigate the diagnostic performance of the automated breast ultrasound system (ABUS), for women 40 years or older for breast cancer, compared to mammography (MG) and hand-held ultrasonography (HHUS). **Participants** = 385 women **Age range** = between 35–67 years old (median 49-year-old) **Variables** = Mammography, Hand-held ultrasonography, Breast cancer, Automated breast ultrasound system	**Screening device** = ➢ ABVS with 6–14 MHz linear broadband transducer ➢ HHUS with 14L5 (5–14 MHz) linear transducer **Duration** = Between July 2016 and December 2016 **Follow up** = 12 months **Interpreting Image** = BI-RADS US lexicon	➢ 75 cases were malignant while 519 were benign or normal. ➢ The sensitivity, specificity, accuracy and Youden index were 97.33%, 89.79%, 90.74% and 0.87 for HHUS; 90.67%, 92.49%, 92.26% and 0.83 for ABUS; 84.00%, 92.87%, 91.75% and 0.77 for MG, respectively. The sensitivity index was 97.33% for HHUS, 90.67% for ABUS, 84.00% for MG; specificity index was 89.79% for HHUS, 92.49% for ABUS, 92.87% for MG; accuracy index was 90.74% for HHUS, 92.26% for ABUS, 91.75% for MG; and the Youden index was 0.87 for HHUS, 0.83 for ABUS, and 0.77 for MG). ➢ Compared to HHUS, ABUS had a significantly superior specificity (*p* = 0.024). ➢ HHUS had highest area under the receiver operating characteristic curve at 0.936, followed by ABUS at 0.916, and MG at 0.884. ➢ The difference was not statistically significant (*p* > 0.05).	✓ ABUS yielded an equivalent diagnostic performance for breast cancer as MG and HHUS, and therefore can be potentially utilized as an alternative technique for the diagnosis of breast cancer.
Niu et al., 2019 China	**Study design** = Prospective study **Objectives** To examine the diagnostic potential of the automated breast ultrasound (ABUS) system in differentiating malignant and benign breast masses in comparison to handheld ultrasound (HHUS). **Participants** = 398 patients **Age range** = Between 29–64 years old, mean age 39 years **Variables** = Handheld ultrasound, Automated breast ultrasound system, Breast cancer, Breast Imaging Reporting and Data System.	**Screening device** = ➢ A 6–14-MHz linear broadband transducer ABUS ➢ A 6–18-MHz linear transducer (18 L6) HHUS **Duration** = between February 2016 to March 2017 **Follow up** = 12 months **Interpreting Image** = BI-RADS US lexicon	➢ Pathological results confirmed 599 masses in total, in 398 women. ➢ 496 of the 599 masses were benign while the remaining 103 were malignant. ➢ No significant differences were found between HHUS and ABUS in terms of positive predictive value (46.46% versus 46.12%), diagnostic accuracy (80.6% versus 80.1%), negative predictive value (95.67% versus 97.96%), and specificity (80.24% versus 77.62%). ➢ Significant differences were found in sensitivity (82.52% versus 92.23%; *p* < 0.01), and in areas under the curve (0.81 versus 0.85; *p* < 0.05) between HHUS and ABUS. ➢ The correlation of the maximum diameter was relatively greater between ABUS and the pathological results (r = 0.885) in comparison to between HHUS and the pathological results (r = 0.855). However, the difference was not significant (*p* > 0.05).	✓ In distinguishing between benign and malignant breast masses, automated breast US is better than HHUS, particularly regarding specificity.
Choi et al., 2018 Korea	**Study design** = Retrospective **Objectives** To examine the hand-held ultrasound (HHUS) compared to the automated breast volume scanner (ABVS), based on the fifth edition of BI-RADS ultrasound **Participants** = 1058 women **Age range** = between 17–79 years old (mean age, 48.2 years) **Variables** = Mammary, Automated breast volume scanner, Hand-held ultrasound, Ultrasonography, Breast imaging reporting and data system, Breast neoplasms.	**Screening device** = ➢ The 15-cm-wide linear array transducer with 5–14 MHz was used for ABVS examinations ➢ The HHUS examinations involved the use of the ACUSON S2000 ultrasound system (or the ACUSON Sequoia 512 **Duration** = Between March 2012 and March 2014 **Follow up** = 12 months **Interpreting Image** = BI-RADS US lexicon	➢ There was moderate to good interobserver agreement in ABVS and HHUS (κ = 0.53–0.67 and 0.55–0.70, respectively), with the exception for associated features for BI-RADS lexicons (κ = 0.31 and 0.36, respectively). ➢ Irregular shape, posterior features (combined or shadowing), and a non-circumscribed margin were individually linked to in both the ABVS and HHUS, malignancy. ➢ The existence of calcification on ABVS (odds ratio (OR), 95% confidence interval (CI): 2.09, 1.11–3.94), and non-parallel orientation on HHUS (OR: 95% CI: 2.04, 1.10–3.78) were individually linked to malignancy. ➢ No significant differences were found between HHUS and ABVS in sensitivity (84.2% vs. 84.2%), specificity (83.9% vs. 80.5%), or AUC (0.90 vs. 0.88).	✓ Based on the fifth BI-RADS edition, there is no statistically significant difference between ABVS and HHUS in terms of diagnostic performance and interobserver variability.
Zhang et al., 2018 China	**Study design** = cohort study design **Objective** To assess the clinical performance of ABUS in comparison to mammography (MG) and the handheld ultrasound (HHUS), for breast cancer detection. **Participants** = 1973 patients **Mean age** = 45.4 ± 9.7 years **Variables** = Automated breast ultrasound, Hand-held ultrasound, Breast Imaging Reporting and Data System, Breast neoplasms	**Screening device** = ➢ ABVS with 6–14 MHz linear broadband transducer. ➢ HHUS was performed using the GE LOGIQ9, Aixplorer system, iU22 Ultrasound System and s2000. ➢ MG images included Fujifilm FDR MS-2500, Hologic Selenia, GE Sengraphe DS, dan. **Duration** = Between February 2016 and March 2017 **Follow up** = NP **Interpreting Image** = BI-RADS US lexicon	➢ 620 (31.4%) and 1,353 (68.6%) of these were classified as BI-RADS categories 4–5 and 1–3, respectively. ➢ The Kappa value and the agreement rate between the ABUS and HHUS were 0.860 (*p* < 0.001) and 94.0%, respectively; and between the ABUS and MG they were 0.735 (*p* < 0.001) and 89.2%, respectively. ➢ In terms of consistency between the results of pathology and imaging, 78.6% of women classified as BI-RADS 4–5 using ABUS later were diagnosed as having cancer or precancerous lesions. This was 7.2% higher compared to women classified using HHUS. ➢ The false-negative rates of HHUS and ABUS for BI-RADS 1–2 was much lower than those of MG and were nearly identical.	✓ A good diagnostic reliability was observed for ABUS. ABUS is thus a promising alternative in detecting breast cancer in China due to its lower dependence on the operator and its performance in detecting breast cancer in women with high-density breasts.
Schmachtenberg et al., 2017 Germany	**Study design** = prospective study **Objective** To determine the diagnostic value of the automated breast volume scanning (ABVS) in comparison to the handheld ultrasonography (HHUS) by using the breast magnetic resonance imaging (MRI) as the gold standard. **Participants** = 28 women **Age range** = between 26–76 years old (mean age 44.6 years) **Variables** = BI-RADS, Automated breast volume scanner (ABVS), Ultrasonography, Breast lesions	**Screening device** = ➢ ABVS with −14 MHz linear broadband transducer. ➢ HHUS with 14L5 (5–14 MHz) linear transducer ➢ MRI **Duration**= Between July 2016 and December 2016 **Follow up** = NP **Interpreting Image** = BI-RADS US lexicon	➢ HHUS detected 54 lesions, MRI detected 72 lesions, and ABVS detected 59 lesions. ➢ No significant difference was found between HHUS and ABVS regarding sensitivity (100% vs. 93.3%), specificity (83.3% vs. 83.3%), diagnostic accuracy (89.7% vs. 87.2%), positive predictive value (78.9% vs. 77.8%), and negative predictive value (100% vs. 95.2%). ➢ In terms of lesion localization (same quadrant), the agreement was 91.2% for MRI and HHUS, and 94.3% for MRI and ABVS. ➢ The assessment of size of lesion was (+/−3 mm) correct in 80% (ABVS) and 79.4% (HHUS) compared to MRI lesion size. ➢ The correlation of measurement of size was moderately higher for ABVS-MRI (r = 0.89) than for HHUS-MRI (r = 0.82), with *p* < 0.001.	✓ ABVS is a probable option to HHUS. ✓ Although ABVS has limitations in assessing the axillary lymph nodes, and is lacking in elastography or Doppler capacities, which occasionally give significant additional information in HHUS, ABVS has the advantages of better reproducibility and operator independence.
Hellgren et al., 2016 Sweden	**Study design** = Retrospective study **Objective** To compare the specificity and sensitivity of ABVS with the handheld breast US in detecting breast cancer under the situation of recall post mammography screening. **Participants** = 180 women **Age range** = aged between 40–75 years old (mean 55.6 years) **Variables** = Mammography, Breast, Primary neoplasms, Ultrasound	**Screening device** = ➢ ABVS with 14 MHz frequency. ➢ HHUS using a linear L17-5 transducer or a L12-5. **Duration** = 2 months **Follow up** = 12 to 24 months **Interpreting Image** = BI-RADS US lexicon	➢ Twenty-six cancers were discovered in 25 women. ➢ Both ABVS and handheld US, used for suspicious mammographic finding in breasts (*n* = 118), yielded the sensitivity of 88% (22/25). The specificity of the handheld US was 93.5% (87/93) while it was 89.2% (83/93) for the ABVS. ➢ ABVS and the handheld US, used in breasts with negative mammography (n¼103), yielded the sensitivity of 100% (1/1). ➢ The specificity of the ABVS was 94.1% (96/102) while the specificity was 100% (102/102) for the handheld US.	✓ The ABVS has the potential to replace the handheld US for the investigation of women recalled from mammography screening due to dubious mammographic findings.
Kim et al., 2016 Korea	**Study design** = prospective study **Objective** To compare the diagnostic performance of the automated breast volume scanner (ABVS) and the handheld ultrasound (US) as a second-look US techniques subsequent to preoperative breast magnetic resonance imaging (MRI) **Participants** = 40 women **Age range** = NP **Variables** = Breast cancer, Breast ultrasound, Second look ultrasound, Magnetic resonance imaging.	**Screening device** = ➢ MRI scan. ➢ ABVS imaging that was performed contained a 5–14-MHz wide-aperture linear transducer. ➢ HHUS with 7–15-MHz and 6–14-MHz linear array transducers. **Duration** = Between 1 March 2014, and 30 September 2014 **Follow up** = 12 months **Interpreting Image** = BI-RADS US lexicon	➢ For the second-look examination, the ABVS has a higher detection rate compared to the handheld US (94.7% vs. 86.8%; *p* < 0.05). ➢ Out of 76 lesions in total, only 1 was discovered by the handheld US, only 7 were identified by the ABVS, while neither the handheld US nor the ABVS could detect the 3 lesions. ➢ Both the handheld US and the ABVS had a lower ability in detecting non-mass lesions compared to the ability in detecting mass-type lesions (*p* < 0.05).	✓ As a method for pre-operational assessment of breast cancer patients, ABVS has a higher efficiency compared to handheld US for a second-look US examination subsequent to preoperative breast MRI. ✓ In non-mass lesion detection, both techniques have limitations.
Jeh et al., 2015 Korea	**Study design** = Prospective study **Objective** To compare the clinical utility of HHUS and ABUS in breast lesion diagnosis and detection. **Participants** = 173 women **Age range** = between 20–80 years old (mean age, 48 years) **Variables** = Mammary, Ultrasonography, Early detection of cancer, Breast, Diagnosis.	**Screening device** = ➢ ABVS with a 5–14 MHz wide-aperture linear probe. ➢ HHUS using a 7–15 MHz and a 6–14 MHz linear transducer. **Duration** = Between March and August 2012 **Follow up** = NP **Interpreting Image** = BI-RADS US lexicon	ABUS overall detection rate was 83.0% while HHUS overall detection rate was 94.2%. ➢ Ten lesions, of which were microcalcifications (nine benign lesions and one malignant), could neither be detected by ABUS nor HHUS. ➢ Out of 194 lesions detected by HHUS, ABUS detected 169 while the other 25 benign lesions were left undetected. ABUS could less frequently detect smaller-sized lesions, including those of lower final-assessment category and benign appearance (*p* < 0.0001 and *p* = 0.011, respectively).	✓ The detection of all malignant lesions by HHUS were similarly made by ABUS. ✓ ABUS failed to detect a few smaller benign lesions.
Choi et al., 2014 Korea	**Study design** = Retrospective study **Objective** To examine whether the estimation of cancer detection is influenced by different ultrasound systems. **Participants** = 1866 ABVS and 3700 HHUS participants **Age range** = between 19–82 years old, mean age ± SD: 47 ± 9 years **Variables** = Screening, Breast cancer, ultrasound, Automated breast volume scanning	**Screening device** = ➢ ABVS with 5–14 MHz with a 9 MHz centre frequency. ➢ HHUS with a bandwidth of 5–12 MHz and a linear array transducer. **Duration** = Between September 2010 to August 2011 **Follow up** = 6 months to 1 year **Interpreting Image** = BI-RADS US lexicon	➢ ABVS had a recall rate of 2.57 per 1000 (48/1866) while HHUS was 3.57 per 1000 (132/3700), with a substantial difference (*p* = 0.048). ➢ Cancer detection yield was 3.8 per 1000 for ABVS, and 2.7 per 1000 for HHUS. ➢ The diagnostic accuracy, with a statistical significance of (*p* = 0.018), was 96.5% for HHUS and 97.7% for ABVS. ➢ The specificity of HHUS and ABVS were 96.7% and 97.8%, respectively (*p* = 0.022).	✓ The performance of ABVS is as good as HHUS in detecting lesions. ✓ ABVS could lead to greater anxiety for the women due to more follow-ups since it has a lower positive predictive value and higher recall rate. ✓ If given the option, over 50% of the women would prefer HHUS.
Kim et al., 2013 Korea	**Study design** = Retrospective study **Objectives** To compare between the automated whole breast ultrasound (AWUS) and the hand-held breast ultrasound (HHUS) in terms of their detection performance. To evaluate the variability of interobserver in interpreting AWUS. **Participants** = 45 women **Age range** = NP **Variables** = Observer variation, Ultrasonography, Breast	**Screening device** = ➢ Mammograms. ➢ ABVS with a 5–14 MHz frequency. ➢ HHUS using a 7–15 MHz linear probe and a 6–14 MHz linear probe. **Duration**= From October of 2009 to March of 2010 **Follow up** = NP **Interpreting Image** = BI-RADS US lexicon	➢ The malignancy detection rate for HHUS was 98.0%, while that of the three readers for AWUS were 90.0%, 88.0% and 96.0%. ➢ In HHUS, the specificity and sensitivity were 62.5% and 98.0%, 90.0% and 87.5% for reader 1, 88.0% and 81.3% for reader 2, and 96.0% and 93.8% for reader 3, in AWUS ➢ No significant difference was found in the sensitivity, specificity, and detection performance of the radiologists (*p* > 0.05) between the two modalities. ➢ Fair to good interobserver agreement was found for size, location of breast masses, ultrasonographic features, and categorization.	✓ AWUS is assumed as beneficial for breast lesion detection. ✓ There is no significant difference between HHUS and AWUS in terms of specificity, detection rate, and sensitivity. However, AWUS demonstrated high degree of interobserver agreement.
Chen et al., 2013 China	**Study design** = Retrospective study **Objective** To clarify ABVS value compared to HHUS in the differentiation of malignant and benign breast masses **Participants** = 182 women **Age range** = Between 16–71 years old; mean age, 41.7 years **Variables** = Automated breast volume scanner, Ultrasonography, and Breast cancer.	**Screening device** = ➢ ABVS with integrated (15.4 cm) 14L5BV linear transducer ➢ HHUS with 9L4 (4–9 MHz) or 14L5 (5–14 MHz) linear transducers. **Duration** = September 2010 and April 2012 **Follow up** = NP **Interpreting Image** = BI-RADS US lexicon	➢ No differences were shown between the ABVS and HHUS regarding accuracy (88.1% vs. 87.2%), specificity (86.2% vs. 87.5%), sensitivity (92.5% vs. 88.0%), false-positive rate (13.8% vs. 12.5%), false-negative rate (11.8% vs. 7.5%), negative predictive value (96.3% vs. 94.3%), and positive predictive value (74.7% vs. 75.6%) (*p* = 0.05 for all).	✓ There is no difference in diagnostic accuracy between HHUS and ABVS in discriminating malignant or benign breast masses.
Lin et al., 2012 China	**Study design** = Prospective study **Objective** To examine the clinical utility of automated breast volume scanner (ABVS) compared to the handheld ultrasound for detecting and providing diagnosis of breast lesions. **Participants** = 81 women **Age range** = between 16–78 years old; mean 40.7 years **Variables** = Automated, Ultrasonography, Breast lesion, Diagnostic imaging	**Screening device** = ➢ ABVS with a wide aperture 14L5BV linear array transducer and central frequency of transducer varied from 9 to 11 MHz. ➢ HHUS that uses the 18L6HD linear array transducer at 10 MHz grayscale central frequency. **Duration** = 1 month **Follow up** = 12 months **Interpreting Image** = BI-RADS US lexicon	➢ 95 breast lesions were detected by both automated breast volume scanner and handheld ultrasound. ➢ Both handheld ultrasound and ABVS demonstrated high specificity (85.0%, and 95.0% respectively), and high sensitivity (both 100%). ➢ For breast neoplasms, a higher diagnostic accuracy was demonstrated by handheld ultrasound (91.4%) than ABVS (97.1%). ➢ No significant difference was demonstrated in maximum diameters of 2D, ABVS, and pathology (*p* > 0.05). ➢ There was better correlation with pathology (r = 0.616) than 2D (r = 0.468) for ABVS.	✓ The automated breast volume scanner is a promising modality in breast imaging with the benefits of operator-independence, high diagnostic accuracy, whole-breast visualization, and greater prediction of lesion size.
Wang et al., 2012 China	**Study design** = prospective study **Objective** To assess the diagnostic value for the discrimination of breast lesions that are benign and malignant, between the automated breast volume scanning (ABVS) and the conventional handheld ultrasonography (HHUS). **Participants** = 213 women **Age range** = aged between 11–81 years, average 43.0 ± 12.5 years **Variables** = Breast sonography, Automated breast volume scanner, Breast lesions, 3D imaging	**Screening device** = ➢ ABVS with frequency of 5–14 MHz transducer (15.4 cm). ➢ HHUS with 18L6 linear transducer. **Duration**= Between August 2010 and December 2010 **Follow up** = NP **Interpreting Image** = BI-RADS US lexicon	➢ The pathology of 239 breast lesions revealed that 154 (64.4%) were benign and 85 (35.6%) were malignant lesions. ➢ There are similarities between ABVS and HHUS regarding specificity (80.5% vs. 82.5%), sensitivity (95.3% vs. 90.6%), accuracy (85.8% vs. 85.3%), positive predictive value (73.0% vs. 74.0%), and negative predictive value (93.3% vs. 94.1%) ➢ Only minor differences were demonstrated by the area under the receiver operating characteristic (ROC) curve used to estimate the accuracy between HHUS and ABVS (0.928 and 0.948, respectively).	✓ In the differentiation of breast lesions that are malignant and benign, HHUS and ABVS had almost identical diagnostic accuracy. ✓ ABVS may assist in uncovering small lesions, demonstrating the presence of intraductal lesions, and distinguishing between real lesions and inhomogeneous areas. ✓ This technique is feasible for clinical application and is a promising new modality in breast imaging.
Wojcinski et al., 2011 Germany	**Study design** = Cohort study design **Objective** To assess the detectability of breast lesions by independent examiner using only ABVS data of breast lesions that were previously detected using conventional ultrasound. **Participants** = 50 women **Age range** = Between 32–72 years old; median, 52 years **Variables** = Automated breast volume scanner, Automated breast ultrasound, ABVS, Breast cancer	**Screening device** = ➢ ABVS with 14L5BV linear transducer (14 MHz, 15.4 cm). ➢ HHUS using 18L6 HD linear transducer (5.5–18 MHz). **Duration** = Between March 2010 and May 2010 **Follow up** = NP **Interpreting Image** = BI-RADS US lexicon	➢ The experimental ABVS yielded a sensitivity of 100% in the described setting (95% CI: 73.2–100%), and 66.0% diagnostic accuracy (95% confidence interval (CI: 52.9–79.1). ➢ The specificity was maintained at 52.8% (95% CI: 35.7–69.2). ➢ Comparison between the ABVS concordance with the gold standard (conventional handheld ultrasound) indicated a fair agreement between both, with the value of Cohen Kappa as an estimation of the inter-rater reliability of κ = 0.37.	✓ For breast ultrasound, ABVS must continue to be perceived as an experimental technique, thus requiring further evaluation studies.
Chang et al., 2011 Korea	**Study design** = Retrospective study **Objective** To examine, retrospectively the detection performance of malignant and benign breast masses with the utilization of 3D volume data obtained by ABUS, and to determine the variables of lesion that influence detectability **Participants** = 67 women **Age range** = Between 20–79 years old, mean age, 47 ± 14 years **Variables** = Detection performance, Screen US, Hand-held ultrasound, Automated breast US.	**Screening device** = ➢ ABVS with 7.5–10 MHz transducer. ➢ HHUS with 6–14 MHz linear transducer. **Duration**= Between November to December of 2007 **Follow up** = minimum 30 months **Interpreting Image** = BI-RADS US lexicon	➢ The sensitivities for the detection of benign and malignant masses were 65.2% (30/46), 95.8% (23/24) for reader 1 (*p* = 0.007), 66.7% (31/46), 87.5% (21/24) for reader 2 (*p* = 0.087), and 56.3% (24/46), 91.7% (22/24) for reader 3 (*p* = 0.001), respectively. ➢ It is demonstrated in the logistic analysis that mass shape (odds ratio, 95% CI; 3.12, 1.02–9.55), size of the mass (odds ratio, 95% CI; 1.12, 1.02–1.24), and changes in surrounding tissue (odds ratio, 95% CI; 0.11, 0.02–0.47) were the variables linked to the ABUS detectability.	✓ Significantly higher sensitivity was demonstrated in reader studies using ABUS data for breast masses that are malignant than benign.
Shin et al., 2011 Korea	**Study design** = Prospective study **Objective** To evaluate, prospectively, the interobserver agreement of five radiologists in detecting lesions, and the characterization in the review of automated ultrasound breast images. **Participants** = 55 women **Age range** = aged between 29–69 years; mean 48 years **Variables** = Automated ultrasound, Breast, Neoplasm	**Screening device** = ➢ ABVS using a 5–14 MHz with a 9-MHz centre frequency. ➢ HHUS using a linear 5–12 MHz transducer. **Duration** = From August to October 2009 **Follow up** = NP **Interpreting Image** = BI-RADS US lexicon	➢ Using the automated ultrasound, the identification of 145 lesions were made by a minimum of two observers. ➢ Lesions larger than 1.2 cm had a detection rate of 92%. ➢ Majority of lesions detected using only handheld ultrasound (11/12, 92%) or automated ultrasound (34/36, 94%) were probably benign masses or cysts. ➢ There is high reliability since all intraclass correlation coefficients for size and location of lesion exceeded 0.75.	✓ The reporting of lesion location and size yielded high reliability. ✓ The description of key features and final assessment category demonstrated substantial agreement.

## Data Availability

Data is available from the studies included in the review that have been cited.

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
