# Peer review of "Evaluation of Diagnostic Performance of Automatic Breast Volume Scanner Compared to Handheld Ultrasound on Different Breast Lesions: A Systematic Review"

_diagnostics, 2022, doi:10.3390/diagnostics12020541_

Round 1

Reviewer 1 Report

In this review, authors compare and found similar diagnostic performance between the automatic breast volume scanner and the handheld ultrasound in the differential diagnosis of benign and malignant breast lesions. This is a well conducted systematic review written in a good level of English language. It highlights the accumulated evidence in this new imaging technology. Unfortunately, there are not many prospective randomized studies comparing this new technology with the handheld US. Till that evidence becomes available; this study will guide the radiologists and breast health care providers. Only concern would be that the researchers included only studies written in English which can cause a Language bias. However, this is commonly seen in the literature and I do not think it would affect the results significantly. I believe this study should be accepted for publication.

Author Response

English language and spell have been checked 

Reviewer 2 Report

This paper constitutes an attempt by the authors to shed light on an important chapter regarding the best modality of mammography screening by comparing two different techniques such as hand-guided ultrasound and Automatic Breast. volume Scanner. I have read with great attention and pleasure, and I must admit that it lends itself well to reading and it is a discursive paper. I also believe that the PRISMA criteria have been respected and that the levels of evidence of the various bibliographic items are also correctly applied. In the Discussion section the Authors not only actively discuss about the papers included in the systematic review, but try to outline different points of view to address the problem. I believe there are minor changes that need to be made before I can accept this job on Diagnostics.
1. Register this systematic review in databases like Prospero.
2.
Please, standardize the style of references. Put the number in square brackets and not the author's name. Thank youLine 65: “morphology of lesions, and” please, make sure the meaning is maintained.

Author Response

Point 1: 1. Register this systematic review in databases like Prospero.

Response 1: I tried to register the work under Prospero but it wasn`t accepted so please is it possible to suggest another database that accepts work after data extracting instead of Prospero.

Point 2: 2. Please, standardize the style of references. Put the number in square brackets and not the author's name. Thank youLine 65: “morphology of lesions, and” please, make sure the meaning is maintained.

Response 2: the references have been modified and for line 65 the meaning has been modified ( It is safe and sensitive for distinguishing the echo of gland tissue and fat, and is good at defining the boundary and morphology of lesions)

  • English language and spell have been checked
  • the conclusion has been modified